

**Measuring snow water equivalent from common offset GPR records through**
**migration velocity analysis**
James St. Clair[1,2] and W. Steven Holbrook[1]
[1]University of Wyoming, Department of Geology and Geophysics, Laramie, WY,
82071, USA.
[2]University of Idaho, Department of Geological Sciences, Idaho Falls, ID, 83402, USA.





**Abstract**

Many mountainous regions depend on seasonal snowfall for their water resources. Current methods of predicting the availability of water resources rely on the long-term relationship between stream discharge and snow pack monitoring at isolated locations, which are less reliable during abnormal snow years. Ground-penetrating-radar (GPR) has been shown to be an effective tool for measuring snow water equivalent (SWE) because of the close relationship between snow density and radar velocity. However, the standard methods of measuring radar velocity can be time consuming. Here we apply a migration focusing method originally developed for extracting velocity information from diffracted energy observed in zero-offset seismic sections to the problem of estimating radar velocities in seasonal snow from common-offset GPR data. Diffractions are isolated by plane-wave-destruction filtering and the optimal migration velocity is chosen based on the varimax norm of the migrated image. We then use the radar velocity to estimate snow density, depth, and SWE. The GPR derived SWE estimates are within 3% of manual SWE measurements when the GPR antenna is coupled to the snow surface and 18% of the manual measurements when the antenna is mounted on the front of a snowmobile ~0.5 meters above the snow surface.



## 1. Introduction

Many regions of the world are critically dependent on seasonal snowfall for their water resources and accurate estimates of how much water is stored in the mountains are necessary to manage this resource. In the United States, there is currently a large network of SNOTEL sites, where automated sensors provide continuous information about snow depth, density, and snow water equivalent that are used to make water availability predictions (Serreze et. al., 1999). While these sites provide valuable information at the site, scaling these point measurements up for basin or grid scale estimates can be challenging (Molotch and Bales, 2005). Currently, these data are used to develop empirical relationships between SWE and nearby stream discharge. These predictions are most accurate during average years and may be not reliable during abnormal years (Bales et al., 2006), thus there is a need to develop new and reliable methods for estimating SWE at a basin scale.

Several previous studies have demonstrated that Ground-Penetrating-Radar (GPR) can be used to measure SWE (e.g. Bradford et al., 2009, Tiuri et al., 1984, Holbrook et al. 2016). Tiuri et al. (1984) showed that at microwave frequencies, the real part of the dielectric constant for dry snow, which governs the velocity, is almost completely determined by the bulk density of snow. However, when liquid water is present, both the real and imaginary parts are needed to determine the volumetric water content of the snow. The complex dielectric constant can be measured by analyzing both the velocity and attenuation characteristics of the snow (Bradford at al., 2009). In the simplest case of dry snow, bulk density can be estimated directly from radar velocity. Snow depth can be measured from the two-



way travel time of the radar pulse between the snow surface and the ground surface
and the velocity. SWE can then be calculated as the product of snow density and
snow height.

Velocity measurements can be made from the surface in several ways.

Common-midpoint gathers (CMP), where the distance between transmitting and
receiving antennas is steadily increased about a central location, provide highly
accurate measurements; the two-way travel-time to subsurface reflectors increases
as a function of offset and velocity. Collecting CMP's requires separable antennas
and it can be time consuming to both collect and process these data. Common-offset
antennas, where both the transmitting and receiving antennas are housed in the
same unit at a fixed offset, allow large amounts of data to be collected with minimal
effort. Measuring the velocity from common offset data can be done in several ways
including calibration from measured snow depths, modeling diffraction hyperbolae
travel-times, and migration focusing analysis.

In this paper, we apply the migration focusing analysis, or migration velocity

analysis (MVA) presented by Fomel (2007) to the problem of estimating radar
velocities, and thus snow density and SWE from 500 MHz common offset GPR
images. After testing the method on two synthetic data sets, we then estimate SWE
from two field data sets. The first data set was collected by pulling the GPR along the
snow surface and the second data set was collected with the GPR antenna mounted
on the front of a snowmobile. Compared to manual SWE measurements, the GPR
derived estimates agree with manual measurements with the estimated
uncertainties.



## 2. Methods


GPR surveys utilize high frequency, broadband electromagnetic signals. The
signal is generated at the transmitting antenna and propagates in three dimensions
at velocity given by $v = c/\sqrt{\kappa'}$, where c is the speed of light in a vacuum and $\kappa'$ is the
real part of the dielectric constant. Signal attenuation is frequency dependent and
can be approximates as $\alpha \approx \sqrt{\frac{\mu_0}{\kappa'}} \frac{\kappa''}{2} \omega$, where $\mu_0$ is the magnetic permeability of free
space and $\kappa''$ is the imaginary component of the dielectric constant (Bradford, 2007).
Both $\kappa'$ and $\kappa''$ are frequency dependent, however within the typical frequency
range utilized for GPR studies only $\kappa''$ exhibits strong variations with frequency; in
dry snow $\kappa'' \approx 0$ (Bradford et al., 2009).
When the GPR signal encounters a boundary between subsurface materials
with contrasting dielectric constants, some of the energy is reflected back and
recorded by a receiving antenna. In this paper, we are specifically interested in
targets that have lateral dimensions that approximate the wavelength of the signal.
These objects appear on the raw GPR image as hyperbolic events, called diffractions,
whose shape depends on the depth of the object and the velocity of the overlying
media. The velocity information contained in diffractions can be extracted by fitting
hyperbolic curves to the data or by migrating the image until the hyperbola is
collapsed to a point or "focus." The latter process is called migration velocity
analysis (MVA). In this paper, we follow an approach described by Fomel (2002)
and develop a semi-automated MVA program in Matlab for the purpose of
measuring radar velocities in seasonal snow. The processing flow consists of three



steps: 1. Separate diffractions from reflections through the process of Plane-Wave-
Destruction (PWD), 2. Migrate the filtered images at a range of potential velocities,
and 3. Use the varimax norm as a measure of diffraction focusing to pick velocities.

**2.1 Data Acquistion**
**2.1.1 GPR data**

During February and March 2015, we collected GPR, snow density, and

snow-depth data in the Medicine Bow Mountains, SE Wyoming. The GPR data were
acquired with a Mala pulse radar system using two common offset antennas with
center frequencies of 500 and 800 MHz. In this paper, we only present 500 MHz
data because the lower frequencies show higher amplitude and more continuous
ground reflections and produces better results when separating reflections from
diffractions.

The GPR data were collected in two ways. In one configuration (Line 19), we

mounted the GPR antenna in a plastic sled and pulled it behind a skier. The unit was
set to fire continuously in time at a rate of 20 traces per second and the sample
interval on each trace was 0.3223 ns. In the other configuration (Line 7) the
antennas were mounted on an aluminum frame attached to the front of a Polaris
RMK 600 snowmobile. The unit was set to fire at a rate of 100 traces per second and
the sample interval was 0.3181 ns. Mounting the GPR antenna in front of the
snowmobile allows us to measure undisturbed snow as well as providing a snow-
surface reflection, which can be used to analyze the attenuation properties of the



snow (Bradford et al., 2009). In both cases, we kept track of our position with a
Trimble R8 GPS unit that recorded our location at 1-second intervals.

**2.1.2 Snow depth and density data**

To validate our snow density and velocity estimates from the GPR data, we

manually measured snow depth and densities.  On Line 7, we used a probe to
measure snow depths at 5-meter intervals along the profile and dug two snow pits;
pit and probe sites were located with a measuring tape.  On Line 19, we dug one
snow pit and located it with a handheld Trimble GPS unit. To measure snow
densities, we used a 0.001 cubic meter, wedge-shaped snow sampler and a scale
that is accurate within 5-10 grams.  We made snow density measurements at 10 cm
intervals in the sidewall of the snow-pits starting from the snow surface and
continuing to the ground.  Pit locations were chosen based on the presence of
diffractions near the snow/ground interface after viewing the GPR images in the
field.

Probed depth measurements are subject to uncertainties due to uneven

ground and deviations in probe angle.  We estimate our depth measurements to be
accurate within +/- 5 cm.  Snow density observations are subject to over and under
sampling and we assign an uncertainty of +/- 5 $g/cm^3$.  We calculate the average
density for each pit profile assigning each snow density observation to a 10 (+/-1)
cm column of snow and performing a weighted sum. Propagating the uncertainties
through the averaging process yields uncertainty estimates of 10-14 % of the



averaged value, consistent with uncertainty estimates for snow pit density
measurements reported by Conger and McClung (2009).

**2.2 Pre-Processing the GPR data**
Prior to MVA we use MATGPR R3 (Tzanis, 2010) to apply several basic
processing steps to the GPR data including: 1. Reset trace to time-zero, 2. Trim time
window, 3. Interpolate traces to equal spacing using the GPS data, 4. Bandpass filter
from 100 to 1000 MHz, and 5. median filter to remove antenna ringing.

**2.3 Plane-Wave-Destruction**
Plane wave destruction (PWD) is a predictive filtering method designed to
suppress events in a seismic or GPR record having a particular dip (Claerbout, 1992;
Fomel, 2002). The GPR image is modeled as the local superposition of plane waves
described by the differential equation (Fomel, 2002):

$$\frac{dP}{dX} - \sigma \frac{dP}{dt} = 0, \qquad (1)$$

where $P(x,t)$ is the wave-field and $\sigma(x,t)$ is the local dip. Equation 1 provides the
means for predicting a trace in the GPR image from its neighbor as a function of
local dip. Fomel's (2002) three-point filter solves this equation:



$$C(\sigma) = \begin{matrix} \frac{(1+\sigma)(2+\sigma)}{12} & -\frac{(1-\sigma)(2-\sigma)}{12} \\ \frac{(2+\sigma)(2-\sigma)}{6} & -\frac{(2+\sigma)(2-\sigma)}{6} \\ \frac{(1-\sigma)(2-\sigma)}{12} & -\frac{(1+\sigma)(2+\sigma)}{12} \end{matrix}, \qquad (2)$$


where $\sigma$ is the local dip and the filtering is accomplished by convolving (2) with the
GPR image. The goal is to suppress continuous reflections that have small dips (such
as snow layering and the ground surface) compared to the steeply dipping
diffraction limbs. Since we do not know the local dips, we use the stencil in equation
2 to estimate them directly from the data.
To estimate local dips, we make an initial guess $\sigma_0$ for the dip (usually zero)
and solve the set of equations

$$\begin{pmatrix} C'(\sigma_0)d \\ \varepsilon D \end{pmatrix} \Delta\sigma = \begin{pmatrix} -C(\sigma_0)d \\ 0 \end{pmatrix} \qquad (3)$$


for$\Delta\sigma$. Here, $C(\sigma)$ denotes the convolution of the filter with the data $(d)$, $C'(\sigma)$ is
the derivative of the filter with respect to $\sigma$ ($C'(\sigma)d$ is a diagonal matrix), $\mathbf{D}$ is the
gradient operator, and $\varepsilon$ is a weighting parameter that controls the smoothness of
the estimated dip field. Imposing smoothness constraints on the dip field estimate
ensures stability in the solution and helps target the reflections in the image, since
they generally show higher amplitudes and are more laterally continuous than the
diffractions we seek to preserve. The estimated dip field is then used to filter the
data.
**2.4 Migration**



Migration is the process that moves reflected and diffracted energy in a
seismic or GPR record to its true location in the subsurface. The quality of the
migration process depends on the accuracy of the velocity estimate. When the
correct migration velocity is chosen, diffraction hyperbolas will collapse to a "focus."
Too low of a velocity and the hyperbola will only be partially collapsed, while a
velocity that is too high will cause the hyperbola to be mapped into a "smile." For the
initial MVA analysis, we migrate the entire image through a suite of velocities (0.19
to 0.29 m/ns) using MATGPR's implementation of the Stolt algorithm (Stolt, 1955).
The Stolt algorithm performs the migration in the frequency wave-number domain
and is computationally efficient.

**2.5 Velocity Picking**
After PWD filtering and migrating the data through the suite of velocities, the
next task is to use a focusing indicator to pick the image that is optimally focused.
Following Fomel (2007), we use the varimax norm (V):

$$V = \frac{N \sum_{i=1}^{N} s_i^4}{\left(\sum_{i=1}^{N} s_i^2\right)^2},$$    (4)

where $s_i$ is the amplitude of the $i$th sample and N is the number of samples included
in the calculation.
V is a measure of the "simplicity" of a signal (Wiggins, 1978). Since the
simplest possible signal is a spike and the optimal migration velocity will map



hyperbolas to the most compact "focus", the maximum V value will correspond to
the image migrated with the optimal velocity.

We choose to compute V within user defined windows, so that we can be

sure to select diffraction hyperbolas that are well preserved after PWD filtering.
After choosing a window, we compute V within this window for each of the migrated
image panels and plot V against migration velocity. Due to noise in the filtered
image and poorly preserved diffractions, the V plot may display multiple peaks.
Plotting the migrated images that correspond to peaks in the V plot allow us to
verify that the diffractions are focused.

After choosing a velocity, we use the shape of the upper portion of the V

curve to estimate uncertainties in the velocity pick. We assume that all velocities
with V values greater than 95% of the peak value could be equally likely, which
yields an upper and lower bound on the velocity estimate that depend on the
sharpness of the V peak. This procedure yields uncertainty estimates of +/- 0.005-
0.01 m/ns, which is comparable to the 0.005 m/ns reported in studies that rely on
picking velocities by visually comparing the migrated images (Bradford and Harper,

2005).

**2.6 Dix Equation**

The migration velocity is the RMS velocity of all of the material between the

GPR antenna and the diffractor. When the GPR antenna is in contact with the snow
and the diffractor is located at the base of the snow, we interpret the migration
velocity to be the average velocity of the snow across the width of the diffraction
hyperbola. When the GPR unit is mounted on the front of the snowmobile, the



signal must pass through the air between the antenna and the snow-surface so that
the migration velocity is higher than that of the snow. To find the snow-velocity
from these data, we use the Dix equation (Dix, 1955):

$$V_{snow} = \left( \frac{V_{mig}^2 t_{soil} - V_{air}^2 t_{snow}}{t_{soil} - t_{snow}} \right),$$ (5)

where velocity subscripts refer to the migration velocity, the velocity in air, and the
velocity within the snowpack and time subscripts refer to the two-way travel-times
of the snow surface and soil surface reflections.
The Dix equation contains two important assumptions. First, the velocity of
the snow must be approximately constant over width of the hyperbola and second,
the half-width of the hyperbola should be small compared to the depth of the
diffractor (x << z). The diffractions in our data sets are approximately 4 to 5 meters
wide, thus we assume that any lateral variations in snow density occur on a larger
scale than this. If the second assumption is not valid, then the Dix velocity will be
higher than the true velocity, resulting in a density estimate that is too low. The
snow depths in our data range from ~1-2 meters, which is comparable to the half-
width of the hyperbolas.
To determine the minimum snow depth that satisfies the x << z assumption,
we traced rays from point diffractors at depths ranging from 0 to 5 meters through a
0.23 m/ns snowpack, representing a snow density of 0.358 g/cm$^3$ (see section 2.7),
with a 0.5 meter thick air layer between the snow surface and the receiver positions
(Figure 1). For each resulting travel-time curve, we obtained nine different



estimates of the migration velocity by performing a least-squares fit to the travel-
time data and successively reducing the widths of the hyperbolas from 10 to 2
meters in 1 meter increments. Using the Dix equation, we obtained estimates of the
snow velocity as a function of diffractor depth and hyperbola width (Figure 2). The
velocity estimates made with the Dix equation approach the true velocity as the
diffractor depth increases and the hyperbola width decreases. For hyperbolas that
are 4 to 5 meters wide (the average width that we observe in our data), the Dix
velocity is within 2 percent of the true velocity when the diffractors are about 1.5
meters deep, 5 percent when the diffractors are about 1 meter deep, and 10 percent
or greater when the diffractors are 0.5 meters deep. We conclude that the use of the
Dix is justified for diffractors buried deeper than 1.5 meters beneath the snow
surface.

Although the results of this analysis are only valid for travel-time modeling,

the x << z assumption may be less severe for migration focusing analysis (see
section 3.1). Diffraction amplitudes decrease with increasing horizontal distance
from the diffractor location, thus the traces closest to the diffractor have the
greatest contribution to the final image, suggesting that the Dix equation may give
adequate results for diffractors that are less than 1.5 meters deep when velocities
are estimated from MVA (we test this with our first synthetic data set in section 3.1).

**2.7 Estimating SWE**

To estimate SWE from the radar data, we need to know the depth of the snow

and the snow density ($SWE = z_{snow}\rho_{snow}$). The depth can be found by picking the





two-way travel-time of the ground reflection and, if applicable, the snow-surface
reflection and then using the velocity estimate to convert time to depth.  Using Eq. 1,
we convert radar velocity to dielectric constant ($v = c/\sqrt{\kappa'}$) and estimate the
density of dry snow with the empirical relationship (Tiuri et al., 1984):

$$\kappa'_d = 1 + 1.7\rho + 0.7\rho^2,$$                                    (6)


where $\kappa'_d$ is the dielectric constant and $\rho$ is the density of dry snow.

In this paper, we assume that our data measure the properties of dry snow

however when liquid water is present in the snowpack the signal attenuates and the
imaginary component of the dielectric constant can no longer be ignored.  Tiuri et al.
(1984) gave the following equation to relate the imaginary dielectric constant of
snow to snow wetness at 1 GHz:

$$\kappa''_s = (0.10W + 0.8W^2)\kappa''_w,$$                                (7)


where W is the volumetric liquid water content, $\kappa''$ is the imaginary component of
the dielectric constant and the subscripts refer to snow and water. $\kappa''_s$ can be
measured by examining the attenuation characteristics of the GPR data (Bradford et
al. 2009) and $\kappa''_w$ can be computed with the Debye relaxation model.







### 2.8 Attenuation analysis


To assess the validity of our dry snow assumption we must estimate the
attenuation properties of the snow. Since the attenuation coefficient increases with
increasing frequency, the higher frequencies attenuate more rapidly than the lower
frequencies. Thus, if there is liquid water present in the snow, it will be manifested
as a reduced frequency content of the base of snow reflection with respect a
reference event that bounds the upper surface of the snow.
To measure the frequency content of the different events in the GPR image,
we compute the local instantaneous frequency attribute (Fomel, 2007). The local
instantaneous frequency is computed in the same way as the instantaneous
frequency except that smoothness constraints are imposed so that the calculations
are less sensitive to noise in the data. We calculate the maximum local
instantaneous frequency within a time window surrounding the event of interest
then average this value across all of the traces in the GPR image. The standard
deviation provides an estimate of the measurement uncertainty.
The soil surface reflection is the obvious choice for measuring the frequency
content of the signal after it has passed through the snow. Our choice of reference
events depends on how the data were collected. When the GPR antenna was
mounted on the front of the snowmobile, we choose the snow surface reflection.
When the GPR was in contact with the snow, we use the arrival that travels directly
from the source antenna to the receiving antenna.
Bradford et al. (2009) gave the following equations to relate the observed
frequency shift to $\kappa''_s$:




$$\frac{1}{Q^*} = \frac{2}{\pi t}\frac{(\omega_0^2 - \omega_t)}{\omega_0^2 \omega_t}, \qquad (8)$$


$$Q^* = \frac{\kappa'_s}{2\kappa''_s}, \qquad (9)$$


where $\omega_0$ is the reference frequency in radians/s, $\omega_t$ is the frequency of the
reflection from the base of the snow, and t is the propagation time of the signal
through the snow. Q* is an empirical constant that assumes that the attenuation
coefficient can be approximated as a linear function of frequency over the
bandwidth of the GPR pulse (Turner and Siggins, 1994). Once we have computed
$\kappa''_s$, we scale the measurement to 1 GHz and use Eq. 7 to estimate W.

At 500 MHz, small changes in frequency result in non-negligible volumetric

water content. Since we expect uncertainties in the frequency measurements of 5-
10 % of the peak frequency (25-50 MHz), it is likely that our data will not allow us to
confidently differentiate between dry snow and moist snow (W=0-0.3) (Figure 3).
**3. Data and Results**
**3.1 Synthetic test**

As a first test on the reliability of migration focusing analysis for

reconstructing radar velocities, we performed the analysis on two synthetic data
sets generated with REFLEX software. The synthetic data sets were generated using
a 500 MHz Kuepper wavelet sampled at 0.0332 ns and traces are 0.1 meters apart.

The first model is 50 meters long and consists of a 0.5 meter thick layer of air

overlying a 0.24 m/ns (corresponding to a density of 0.29 g/cc) layer of snow with
depths that range from 0.5 to 5.7 meters. Beneath the snow is a 0.10 m/ns layer



representative of soil. Along the snow/soil interface there are 16 diffractors buried
at depths ranging from 0.5 to 5.7 meters. The purpose of this data set (Figure 4a)
was to test the performance of the Dix equation on velocities estimated from the
MVA analysis.

After applying the PWD filter, the ground reflection was adequately

suppressed (Figure 4b). We migrated the filtered image at 0.002 m/ns intervals
from 0.18 to 0.28 m/n and measure the optimal migration velocity for each
diffractor by computing V (Figure 4c) within small windows centered over the apex
of the hyperbola (Figure 4b). We use the Dix equation to convert the migration
velocities to the velocity of the snow layer. The average of all snow velocity
measurements is 0.241 m/ns with a standard deviation of 0.04 m/ns.

There is no systematic relationship between the velocities recovered and the

depth of the diffractor (Figure 5). The shallowest diffractor was at ~0.5 m depth and
the recovered velocity was 0.237 m/ns. The greatest differences between recovered
and true velocities were for diffractors at depths of 1.03, 1.54, and 2.1 meters. Here
the recovered velocities were 0.245, 0.246, and 0.245 m/ns. Notably, the peak V
value for the diffractor located at 1.54 meters depth corresponded to an image that
was clearly over-migrated and we would have rejected this measurement for a real
data set.

The second model is 10 meters long with a snow layer that ranges from 1.9

to 2.7 meters thick with a velocity that increases from 0.257 m/ns at x=0, to 0.262
m/ns at x = 10 meters. There are seven diffractors along the soil/snow interface.



The primary purpose of this data set (Figure 6a) was to see whether this method
could resolve a lateral change in velocity.

After applying the same processing flow described above, we recover a

lateral velocity trend that is similar to the true velocity structure (Figure 6b). The
recovered velocities systematically underestimate the true velocities by about 2.1 %
at x = 0 meters, and by ~1.6% at x = 10 meters.

**3.2 Ski-pulled**

Line 19 is a 74 meter long, skier pulled data set collected on February 25,

2015 in below-freezing conditions. The data show an abundance of diffractions
along the snow/ground interface, likely a result of small boulders, and a few isolated
diffractions within the snowpack, most likely small trees or bushes (Figure 7a).
Since the antenna was coupled to the snow, we compare the average frequency of
the direct wave to that of the soil reflection to determine whether there is any liquid
water present in the snowpack. The average frequency of the direct arrival for every
trace in the image is 410 MHz with a standard deviation of 10 MHz and the average
frequency of the soil reflection across the whole line is 457 MHz with a standard
deviation of 42 MHz. The soil reflection appears to have a higher frequency content
than the reference frequency. We infer that there was no liquid water present in the
snow on this day.

Velocities on this line range from 0.23 to 0.25 m/ns with an average

uncertainty of +/- 0.01 m/ns.  Estimated snow-depths range from 1.6 to 1.9 meters
with an average uncertainty of +/- 0.07 m.  Estimated snow densities range from



0.23 to 0.36 g/cc with an average uncertainty of +/- 0.07 g/cc.  Estimated SWE
ranges from 0.3 to 0.5 meters with an average uncertainty of 0.08 meters (Figure 8).

We measured snow density and depth in a pit located at 68 meters along the

profile. The snow pit showed a depth of 1.33 meters and an average density of 0.30
+/- 0.04 g/cc resulting in a SWE measurement of 0.40 +/- 0.07 meters. GPR derived
estimates at the pit location are:  snow depth = 1.28 +/-0.06 meters, density = 0.32
+/- 0.07 g/cc, SWE = 0.41 +/- 0.07 meters.

**3.3 Snowmobile Mounted**

Line 07 was collected on the morning of March 11, 2015 in a flat meadow just

south of Wyoming State Highway 130. This line is 98 meters long and shows an
abundance of diffractions along the snow/ground interface (Figure 9a).  Picking
velocities along this line required significantly more discretion than was required on
Line 19. Whereas on Line 19 we were confident in choosing velocities with well-
defined varimax peaks, on this line we rejected some velocity observations between
x = 0 and x = 10 that appeared to produce well focused diffractions that would have
resulted in snow-density estimates greater than 1 g/cc.

Velocities on this line range from 0.22 to 0.24 m/ns with an average

uncertainty of +/- 0.012 m/ns.  Estimated snow depths range from 0.6 to 1.8 meters
with an average uncertainty of +/- 0.07 m.  Estimated snow densities range from
0.27 to 0.45 g/cc with an average uncertainty of +/- 0.05 g/cc.  Estimated SWE
ranges from 0.26 to 0.8 meters with an average uncertainty of 0.08 meters.





The snowpits located at 50 and 97 meters showed average snow densities of
0.38 and 0.36 g/cc and SWE values of 0.54 and 0.64 meters. The GPR derived SWE
estimates 50 and 97 meters were 0.44 +/-0.08 and 0.74 +/- 0.12 meters. Compared
to the probed snow-depths, the GPR estimated snow-depths are generally low
(Figures 10b and 11) and, on average, within 8% of the probed depths. The
correlation coefficient between predicted and observed snow depths is 0.95.
During data acquisition on Line 07, the air temperature was 5° C and we
expect there to be liquid water present in the snow. The average frequency of the
snow reflection for every trace in the image is 435 MHz with a standard deviation of
27 MHz and the average frequency of the soil reflection across the whole line is 464
MHz with a standard deviation of 38 MHz.  Again, the frequency content of the soil
reflection appears to be higher than the reference frequency. Within the uncertainty
bounds there is no resolvable frequency change, however given these uncertainties
there may be up to a 36 MHz shift, which would result in a volumetric water content
of less than 0.03 (Figure 3).
**4. Discussion**
The primary purpose of this study is to simplify the process of measuring
GRP velocity in seasonal snow and obtain reliable SWE estimates. Common offset
GRP data are fast and easy to obtain and velocity estimates can be made when
diffractions are present. However, the common methods of visually inspecting
migrated images or fitting curves to diffraction hyperbolas can be time consuming
and subject to human error. The migration velocity analysis described in this paper

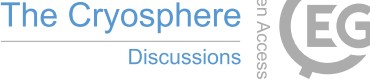

provides an efficient means for extracting velocity information from large GPR data
sets. Here we discuss the performance of this method.

The PWD method of separating continuous reflectors from diffractions treats

the GPR image as the superposition of locally planar waves. Estimating the slope of
these waves from the image requires the solution of a regularized inverse problem
and the smoothness of the slope-field depends on the choice of regularization
parameter. We found that areas of rapidly changing slope can result in noise from
incompletely suppressed reflection events and poorly preserved diffractions. Noise
from inadequate filtering may cause the Varimax Norm value to be high even when
the diffractions are not optimally focused and low when they are.  Visually checking
the migrated images before committing to a velocity pick can help mitigate this
issue.

In particular, Line 7 required a substantial amount of user intervention to

avoid picking obviously incorrect velocities. The performance of the MVA analysis
along this line may have been due to several complicating factors: 1. When mounted
on the snow-mobile, the GPR antenna is fixed at the rear and can wobble up and
down at the front by up to ~5 cm. The change in orientation of the antenna with
respect to subsurface targets as well as the change in distance between the snow
surface and the GPR antenna may be additional noise sources and cause diffractions
to migrate incorrectly. This situation is likely to be of concern when the snow-
surface is uneven, or when the snowmobile is accelerating. Indeed, the greatest
variability along this line occurred during the first few meters when the snowmobile
was accelerating.  2. On this day the air-temperatures were above freezing and,



although our frequency analysis suggests that we can make the dry snow
assumption, it is likely that some water was present in the snowpack the presence of
water in the snowpack would result in decreased velocities and increase the
apparent dry snow density.

The velocity values that we measured, when converted to snow density,

agree with our snow pit density measurements within the uncertainty estimates.
One way to evaluate the efficiency of a model is the Nash Sutcliffe Efficiency (NSE)
coefficient (Nash and Sutcliffe, 1970). The NSE ranges from $-\infty$ to 1 and measures
the quality of predicted values relative to the mean observed values. An NSE of 1
occurs when the predicted values are in perfect agreement with the observations,
negative values indicate that the mean observed value is superior to the predicted
values and 0 suggests that they are equivalent. For Line 7, NSE coefficients are 0.77,
-29.9 and -4.75 for the predicted snow depths, SWE, and densities. Since snow
depths are highly variable and our velocities estimates are reasonably
representative of snow, it is not surprising that our predictions would match the
data better than the mean probe measurement. The negative values for SWE and
density predictions suggest that averaging mean snow-densities from manual
observations may be a better strategy. However, we only have two density
observations to compare the predictions to. We suggest that our method would
work well to make coarse estimates of SWE across large areas with minimal effort,
but if greater accuracy is required a sparse number of manual observations may be
useful to supplement and/or ground truth the GPR estimates.



The data presented in this paper contained an abundance of diffractions
located near the soil/ground interface allowing an average velocity for the entire
snowpack to be obtained.  These events are likely due to the presence of rocks and
small bushes near the base of the snow-pack, which may not be present in all
environments.  Areas likely to contain point diffractors suitable for this type of
analysis can be scouted for ahead of time during the summer months or on aerial
photographs.
**5. Conclusions**
We applied the migration focusing analysis presented in Fomel (2007) to the
problem of estimating SWE in seasonal snow.  The method was most accurate for
the case when the GPR was in contact with the snow when GPR derived SWE
estimates were within 3 % of the manual observation. When the GPR was mounted
on a snowmobile, the results were within 18% of the manual observations.
The processing flow that we presented in this paper proved to be an efficient
way to measure radar velocities within seasonal snow. While not fully automated,
the method requires less processing time than visually scanning each migrated
image and could make GPR a more attractive tool for estimating SWE at the
watershed scale.









**Acknowledgements**

This work was funded by the U. S. National Science Foundation (NSF)

Wyoming EPSCoR Program, NSF award EPS-1208909. We would also like to thank
Matt Provart for assisting with data collection and Mehrez Elwaseif for assistance
with REFLEX software. Data used in this paper are available at
https://data.uwyo.edu.




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



**Figure Captions**

**Figure 1** Raypaths and travel-times for point diffractors. **a)** 0.5 meters of air overlying a 230 m/ns snowpack with point diffractors buried at 0.5 meter intervals. **b)** two-way travel-times for each of the diffractors showing the characteristic hyperbolic shape.

**Figure 2** Dix velocities for point diffractors as a function of depth for different hyperbola widths. The true interval velocity is 0.230 m/ns (red line) and the Dix velocities are shown as black lines. The red dashed line is at 0.234 m/ns, which is 2 percent greater than the true velocity.

**Figure 3** Snow wetness for typical GPR velocities in and snow and peak frequency shifts for a reference frequency of 500 MHz. For the data presented in this paper, typical uncertainties in the frequency measurements are 10 to 40 MHz and the velocities range from 0.22 m/ns to 0.25 m/ns. For the typical range of velocity and frequency shift estimates reported in this paper, snow wetness values less than 0.03 cannot be resolved.

**Figure 4** Synthetic Data set and velocity picking. **a)** synthetic data before filtering. **b)** the unmigrated data after PWD filtering, black box indicates windowed portion of the data used to calculate the Varimax norm. **c)** Varimax norm plotted against velocity showing a peak at 0.246 m/ns. **d)** windowed portion of the data migrated at 0.246 m/ns showing focused diffraction events.




**Figure 5** Velocities from synthetic data set as a function of diffractor depth. Solid
blue line shows measured migration velocities, dashed blue lines show uncertainty
bounds. Solid red line show velocities computed with the Dix equation, dashed red
lines show uncertainty bounds. Solid black line shows the true velocity (0.24 m/ns).
Light gray region indicates where velocities are within 2% of the true velocity and
dark gray region shows where velocities are with 5% of the true velocity.

**Figure 6 a)** Synthetic data set from a model with the lateral velocity trend and no
air layer. **b)** The recovered velocities (black line) show the same trend as the true
model (red) but systematically underestimate the true values by 2.1% at x = 0
meters and 1.6 % at x = 10 meters.

**Figure 7**  Velocity picking Line 19. **a)** unmigrated GPR data.  **b)** unmigrated GPR
data after PWD filtering, black box indicates windowed portion of the image used to
compute the varimax norm **c)** Varimax norm for windowed data as a function of
migration velocity showing a peak at  0.250 m/ns  **d)** windowed portion of the data
migrated at 0.250 m/ns showing the focused diffraction events.

**Figure 8** Line 19 Results. **a)** the radar velocity within the snow along the profile.  **b)**
snow depth (black line) and SWE (blue line) estimates from the GPR data, snow pit
data are shown in red. **c)** snow densities estimated from the GPR data (blue line)
and the density measured in the snow-pit at 68 meters (red).






**Figure 9** Velocity Picking Line 07. **a)** unmigrated GPR data. **b)** unmigrated GPR

data after PWD filtering, black box indicates windowed portion of the image used to

compute the varimax norm **c)** Varimax norm of the windowed data as a function of

velocity showing a peak of 0.254 m/ns **d)** windowed portion of the data migrated at

0.254 m/ns showing the focused diffraction events.

623

**Figure 10** Line 07 results. **a)** radar velocity along the profile. **b)** snow depth (black

line) and SWE (blue line) estimates from the GPR data as well as the probe depths

(red) and the snow pit data at 50 and 97 meters (red). **c)** snow densities estimated

from the GPR data (blue line) and the densities measured in pits at 50 and 97

meters (red).

**Figure 11** Cross-plot of snow depths measured with snow-probe (x-axis) and snow-

depths predicted from GPR data (y-axis). R^2 = 0.95.





.**Figures**

**Figure 1**

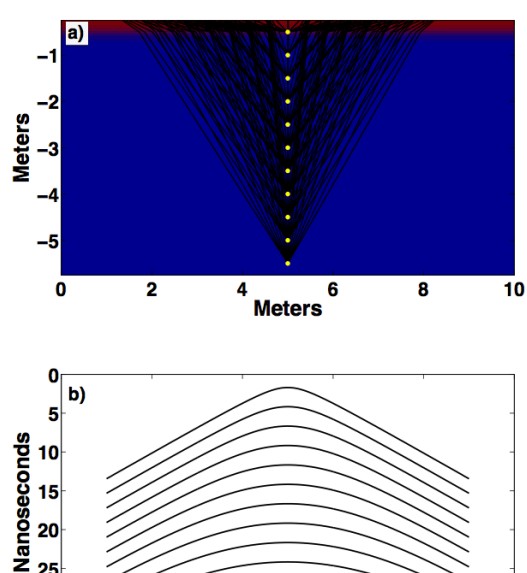




**Figure 2**

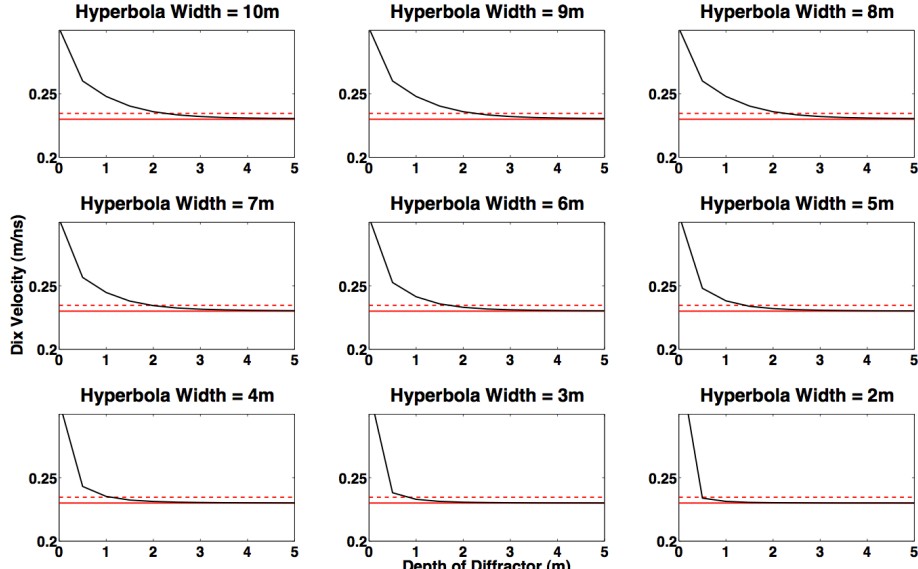



**Figure 3**

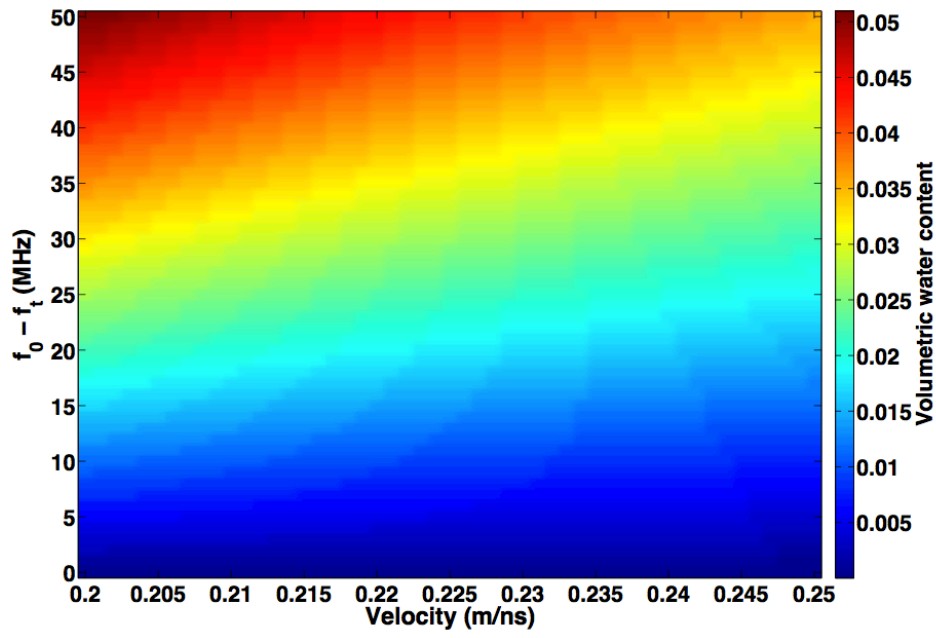





**Figure 4**

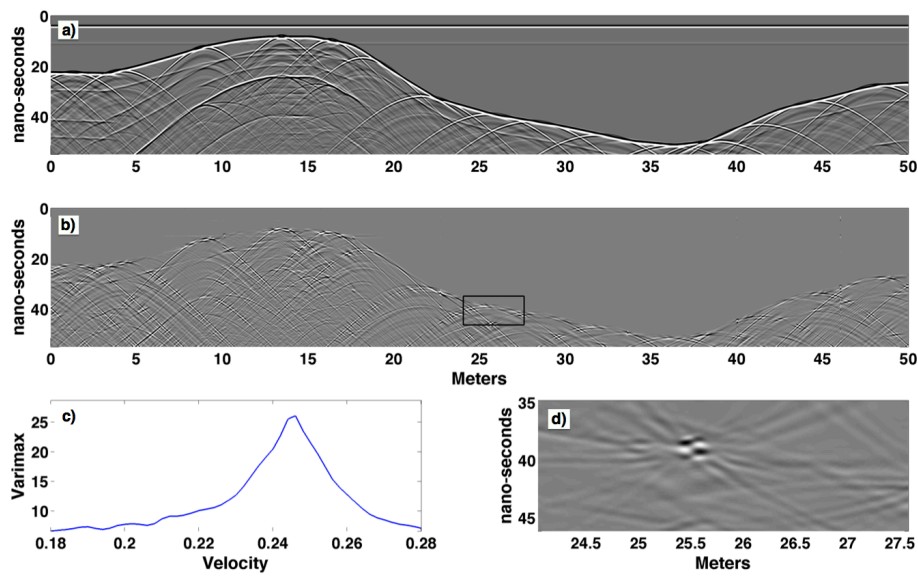





**Figure 5**

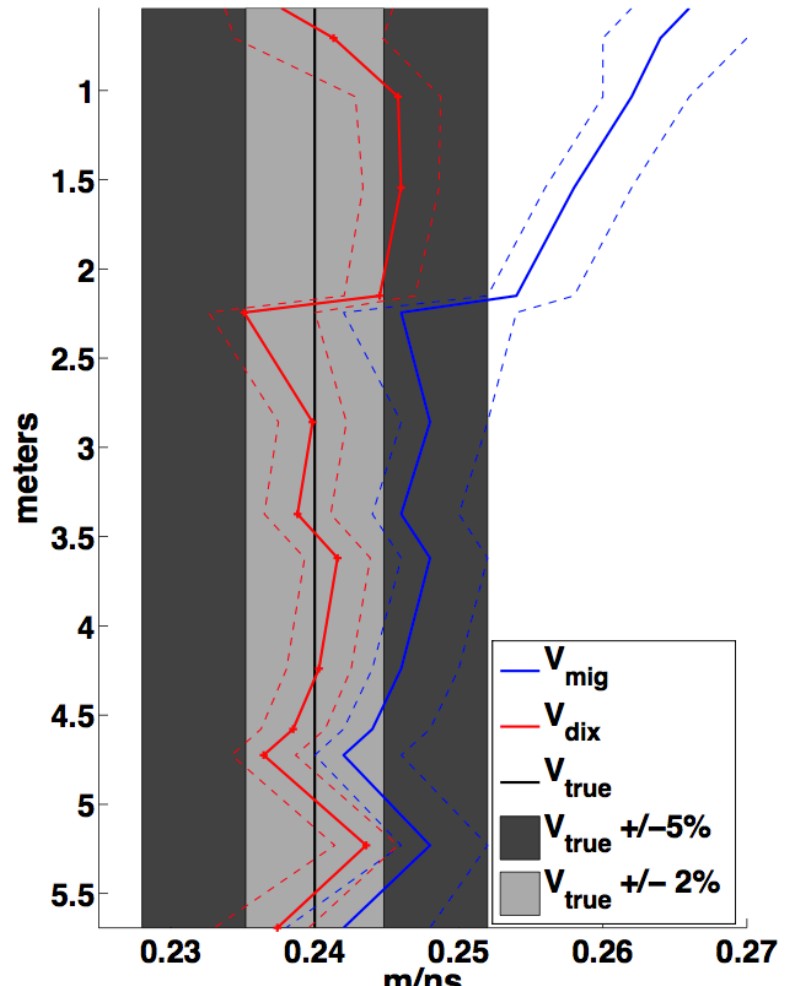



**Figure 6**

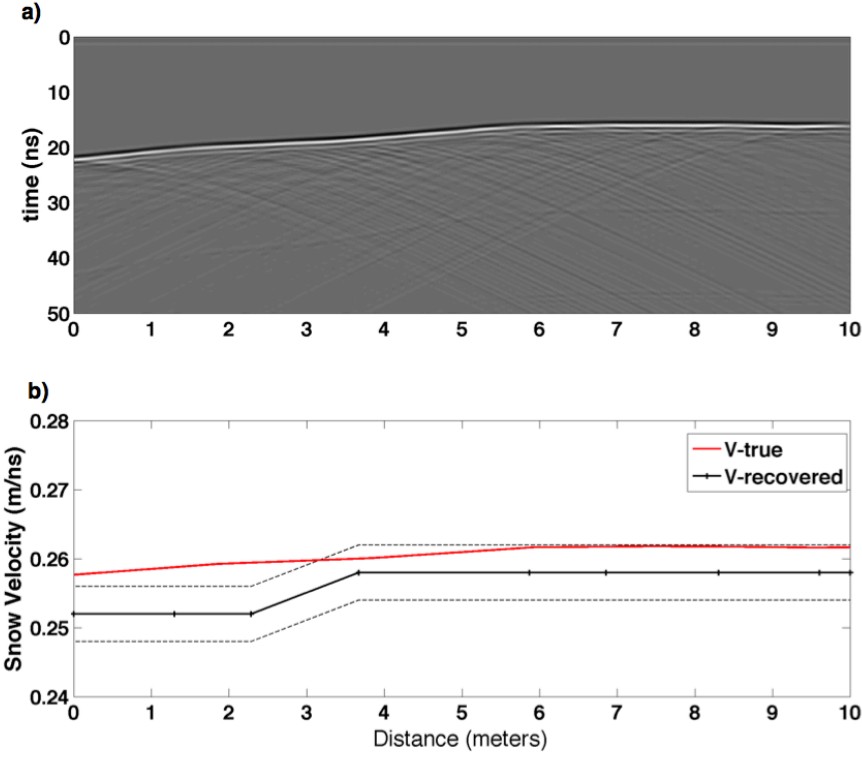



**Figure 7**

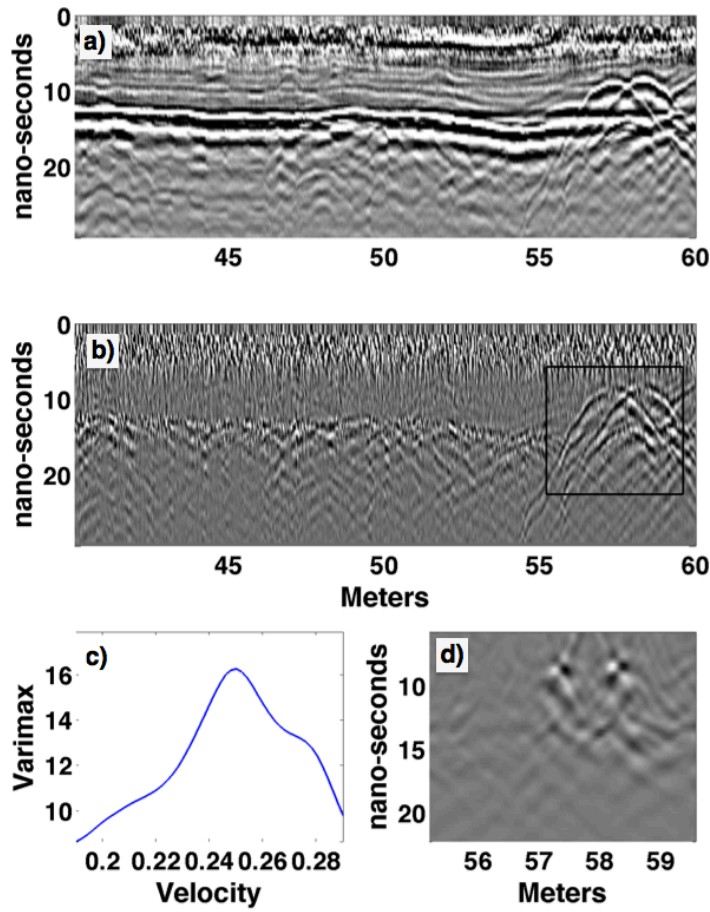





**Figure 8**

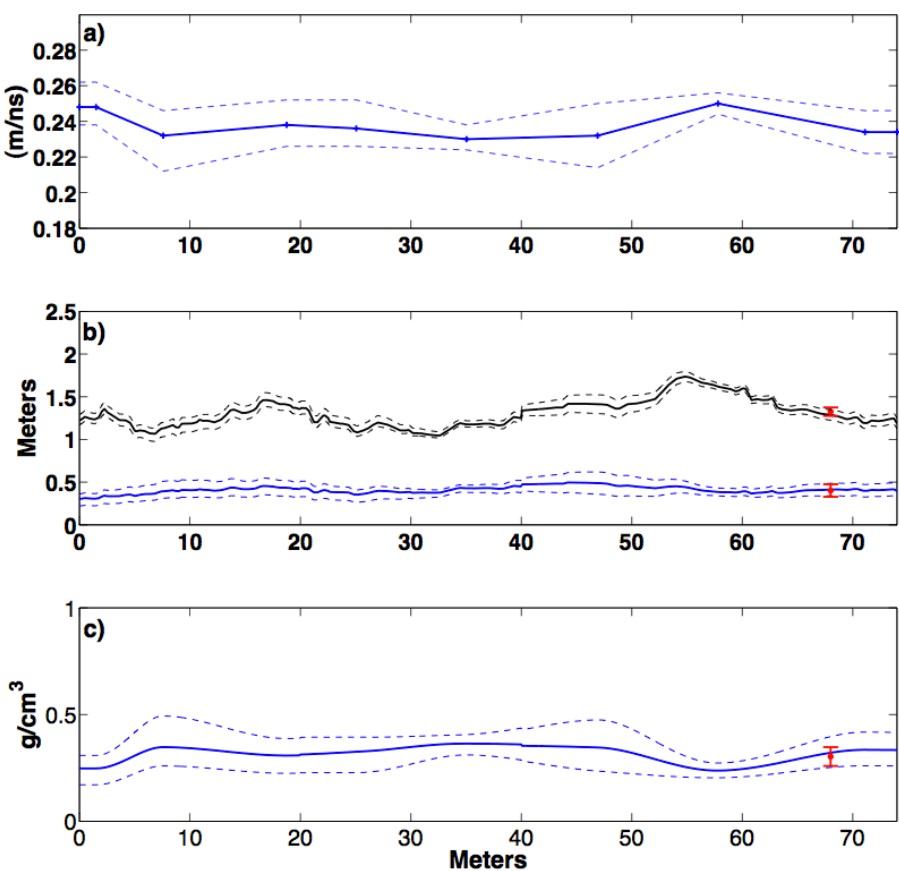




**Figure 9**

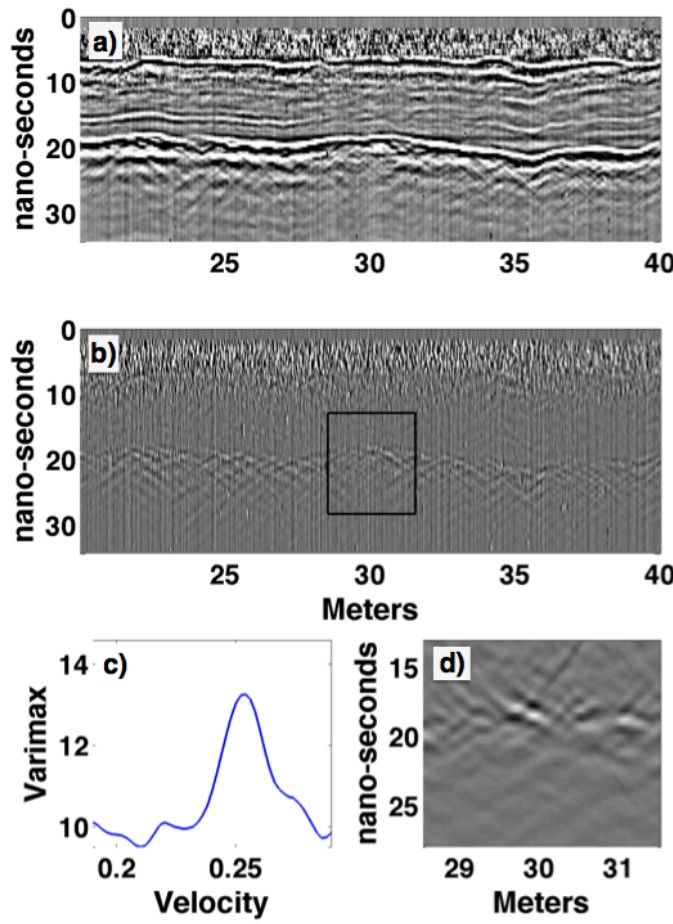





**Figure 10**

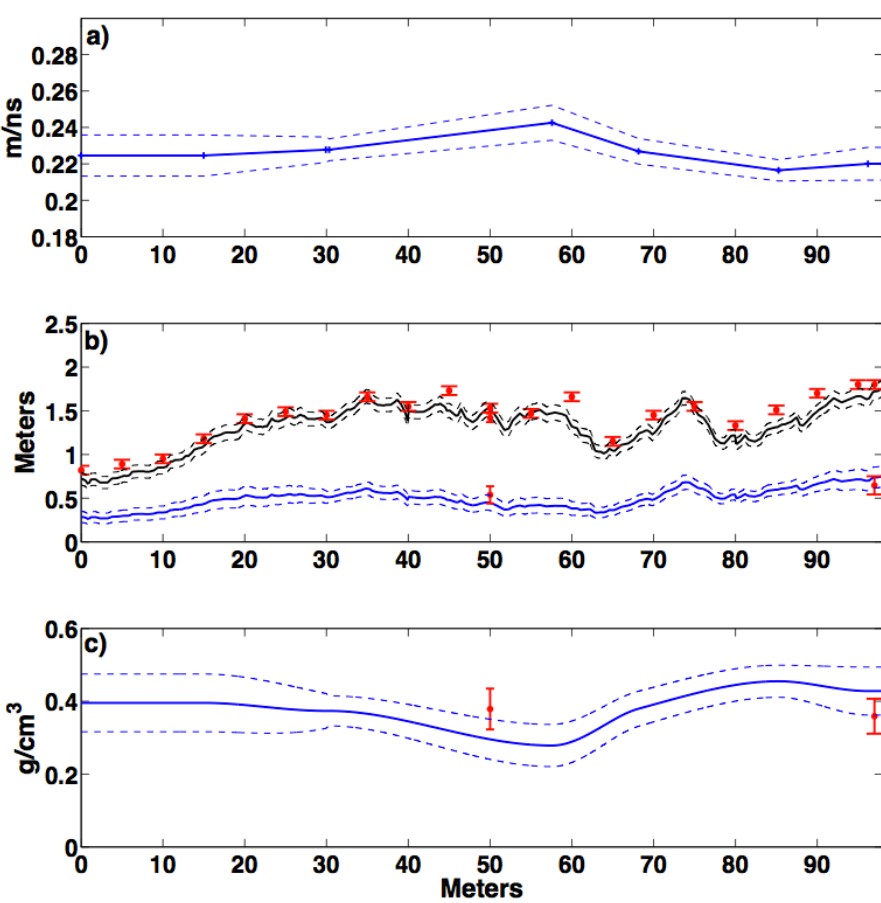




**Figure 11.**

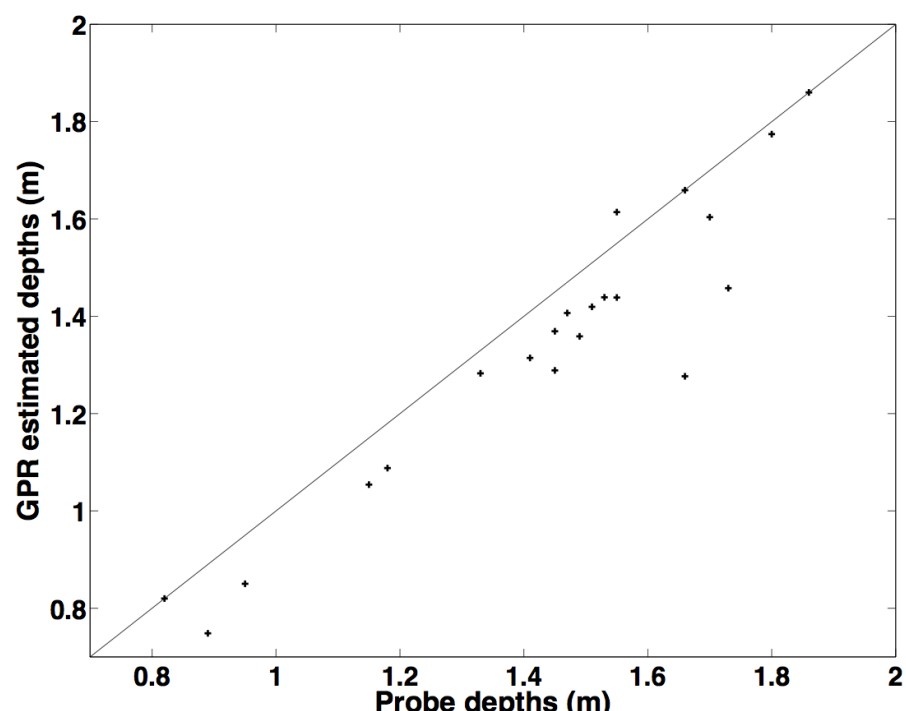