# Peer review of "1. Introduction"

_The Cryosphere, 2017_

## Referee Comment (RC1) · Anonymous Referee #1 · 20 Jun 2017

General Comments:

In this paper, the authors attempt to apply established techniques from the exploration seismology literature to the problem of measuring Snow Water Equivalent (SWE) using constant offset GPR with the stated goal of simplifying that process while obtaining reliable results. The manuscript goes on to detail the application of these complicated methods to a few lines of field data with exceedingly limited success at producing results and processing flows that are either reliable or simple. Given these results (as presented and described in the body of the manuscript itself) the concluding statements that "the processing flow that we presented in this paper proved to be an efficient way

to measure radar velocities within seasonal snow" (Line 508) and "the method requires less processing time than visually scanning each migrated image and could make GPR a more attractive tool for estimating SWE at the watershed scale" (Line 509) are unsupported.

Specific Comments:

The paper states that the "primary purpose of this study is to simplify the process of measuring GPR velocity in seasonal snow and obtain reliable SWE estimates" (Line 444) yet it accomplishes neither:

In terms of processes, simplification, the adaptation of Fomel's seismic approach (Line 133) and the NSE metric (Line 477) are gratuitously complicated, fail due to data quality issues (Line 230), and are not justified given the limited data set. There is no case made that the claims of efficiency (e.g. Line 221, Line 508) address actual measurement and processing bottlenecks or are even accurate. Further, once implemented these techniques still require frequent and substantial manual interventions (Lines 230, 377, 447, 460, and 462) culminating in the authors' suggestion that wavelength scale "point diffractors suitable for this type of analysis can be scouted for ahead of time during summer months or on aerial photographs" (Lines 499). Traditional GPR methods are much simpler than this (e.g. do not require advance scouting or aerial observation).

In terms of reliable SWE estimation, the authors show that the presented approach "does not allow us to confidently differentiate between dry snow and moist snow" (Line 350). Further, authors show that the collected data does not exhibit the frequency dependent attenuation upon which the entire approach depend (Line 318 and Line 331) and even though "within uncertainty bounds there is no resolvable frequency change" (Line 440) the authors go on to use that uncertainty speculate that "there may be up to a 36 MHz shift" and then estimate a volumetric water content from it (Line 442) even though nothing suggests the existence of such a shift beyond the authors' assumption and assertion it should exist (Line 318). This is inappropriate and, as written, likely

invalidates the results and conclusions of the paper. Further, the lack of observed frequency dependence is likely due to an actual lack of frequency dependence in radar attenuation in ice for frequencies below 1 GHz (Gudmandsen, 1971). This must be addressed. At the very least, the authors' statement "averaging mean snow-densities from manual observations may be a better strategy" (Line 489) makes it clear that the stated goal to "obtain reliable SWE estimates" (line 144) was not met.

Technical Corrections:

Line 107: Add an explanation and citation for "targets that have lateral dimensions that approximate the wavelength of the signal".

Lines 122: Why mention the 800 MHz data if it's not used in the analysis? Consider removing.

Line 202: This entire paragraph has a level of detail that seems inappropriate for a journal manuscript. Consider dropping or revising.

Line 235: Provide a justification for the "equally likely" claim.

Line 251: The units of the left and right side of this equation do not match.

Line 318: Add an explanation and citation for "the coefficient increases with increased frequency" and address the fact that, in ice and below 1 GHz it does not (Gudmandsen, 1971).

Line 155, 404, and others: Justify uncertainties and provide basis throughout the manuscript for uncertainties that are currently just asserted.

References:

Gudmandsen, P. "Electromagnetic probing of ice."ÂăElectromagnetic probing in geophysics. Vol. 79. 1971.

---

## Referee Comment (RC2) · H. Maurer (Referee) · 21 Jun 2017

The paper by St. Clair and Holbrook describes a novel approach for determining the snow-water-equivalent (SWE). I have read the well-written paper with great interest, and I judge it a useful contribution for the Cryosphere community. Before the paper can be accepted, I suggest that the authors address the following issues.

1. There are inconsistencies and errors with the units. I suggest that they should use mks units throughout the entire manuscript.

2. It is often referred to Line 7 and Line 19. Without a map showing these profiles

and/or a table describing the characteristics of the profiles, these references are not helpful. Since there seem to be only lines 7 and 19 discussed, it would make sense to rename them to line 1 and line 2.

3. It is my understanding that the methodology works only for dry snow and that determining the liquid water content from the GPR data was not successful. These two facts should be stated more explicitly.

4. Figures 10 and 11 indicate that the GPR estimated depths are systematically smaller compared with the probe depths. This should be discussed.

5. Further discussion is required on the basic assumption of the approach that the scattering bodies can be considered as a point diffractor. I guess that the scattering occurs mostly at sizeable boulders. Therefore, I am not convinced that the point scatterer assumption is really justified. The diffraction hyperbola of a finite sized scattering body is expected to appear wider in a GPR profile, which would likely result in an overestimation of the snow velocity. Consequently, one would thus rather overestimate the snow depth, but, as mentioned in point 4, the opposite seems to be the case.

6. Some of the figures are lacking proper axis labeling.

7. It seems that in Figures 7 and 9 only portions of the GPR profiles are shown. This should be clarified.

---

## Author Comment (AC1) · 6 Sep 2017

We would like to thank the reviewers for their time and remarks on our manuscript "Measuring snow water equivalent from common offset GPR records through migration velocity analysis." Please find the attached file "Author_Response.pdf" it contains our response to the reviewers comments, a revised manuscript and supplementary materials.

Regards, James St. Clair

Please also note the supplement to this comment:

[Figure]

https://www.the-cryosphere-discuss.net/tc-2017-90/tc-2017-90-AC1-supplement.pdf

---

## Author Response (AR1)

We thank the two reviewers for their insightful remarks. In response to these remarks, we have made a number of major changes in the manuscript. We first describe the major changes and then respond to individual comments. The revised manuscript and supplementary material are attached below.

**Major Changes to the manuscript**

1. In our original manuscript, we measured velocities from common offset GPR records by isolating diffractions through Plane Wave Destruction filtering, migrating the filtered data through a suite of constant velocities, and then computing the Varimax norm by isolating individual diffractions or groups of diffractions. This approach required defining the region of interest, viewing the migrated image that corresponded to the peak varimax value, and evaluating the quality of the velocity pick by inspecting the migrated image.

   In our revised manuscript, we compute the varimax norm in sliding windows that encompass the entire time section and have with a fixed width of ~10 meters. The result is a continuous set of velocity estimates along the GPR profile. Spurious picks are suppressed by smoothing the results in x. The revised method does not require inspection of the migrated images.

2. In our original manuscript, we presented two field data sets with independent snow density, depth, and SWE measurements. In the revised we include four additional field data sets with coincident manual observations. These data are included in our validation of the method, results from individual data sets as well as our manual snow density measurements are included in a supplementary materials file.

3. There was some confusion regarding our analysis of radar attenuation our dry snow assumption. Since the main focus of our paper is to estimate the radar velocity and thus the density of dry snow and the attenuation discussion detracts from this focus, we have reduced this discussion to one paragraph and moved it to *Section 2.7 Estimating SWE*.

4. We determined that the second synthetic data set, which was used to test the ability of the method to resolve lateral velocity changes, was redundant and we have removed it from the revised manuscript. Although the remaining synthetic data set has a constant snow velocity, the changing snow depths cause the migration velocity to vary along the profile (Figure 5, revised manuscript).

**Response to reviewer comments:**

Reviewer #1

(Reviewer Comments in italics)

*General Comments:*

*In this paper, the authors attempt to apply established techniques from the exploration seismology literature to the problem of measuring Snow Water Equivalent (SWE) using constant offset GPR with the stated goal of simplifying that process while obtaining reliable results. The manuscript goes on to detail the application of these complicated methods to a few lines of field data with exceedingly limited success at producing results and processing flows that are neither reliable or simple. Given these results (as presented and described in the body of the manuscript itself) the concluding statements that "the processing flow that we presented in this paper proved to be an efficient way to measure radar velocities within seasonal snow" (Line 508) and "the method requires less processing time than visually scanning each migrated image and could make GPR a more attractive tool for estimating SWE at the watershed scale" (Line 509) are un- supported.*

In response to your general, specific and technical comments, we have made a number of major changes in the manuscript.

1. Most significantly, we have eliminated the need for examining isolated diffractions in the GPR image. In the revised manuscript, we now compute the Varimax Norm in ~10-meter-wide sliding windows that span the entire time section. The result is a continuous set of varimax curves along the GPR profile and choosing the peak of each curve provides a continuous estimate of the migration velocity along the profile. Although some of these curves provide spurious velocity estimates, smoothing the velocity picks in the horizontal direction reduces the influence of spurious velocity picks.

2. Our revised manuscript includes the results of six field GPR data sets, four pits, and 86 probed depth observations.

3. To qualitatively asses the efficiency of the method, we include an example of the processing time required for one 100 meter long GPR profile on a 2016 MacBook Pro with a 2 GHz processor. Lines (727-734):

> "One of our main goals was to produce a processing flow that allows for the rapid processing of common offset GPR data with minimal user interaction. The two most time computationally expensive parts of the processes are the migrations and the varimax calculations. As an example, on a 2016 MacBook Pro with a 2GHz processor, for the ~ 100-meter-long Line 4, performing 51 migrations takes approximately 5 minutes, the varimax calculation takes about half as long, and the PWD filtering takes a few seconds. The most time-consuming part of the process is picking the arrival times of snow surface and ground surface reflections."

***Specific Comments:***

*The paper states that the "primary purpose of this study is to simplify the process of measuring GPR velocity in seasonal snow and obtain reliable SWE estimates" (Line 444) yet it accomplishes neither:*

We admit that the word "simplify" was the wrong word to use. The goal of our study is to develop an efficient method for obtaining GPR velocities in seasonal, and thus snow densities, from common-offset GPR data. Additionally, we sought to minimize the amount of human interpretation throughout the process.In the revised manuscript, this sentence reads (lines 664-666):

> "The primary purpose of this study is to develop an efficient processing flow for measuring GPR velocity and thus snow density SWE from common-offset data that requires a minimum amount of human interpretation."

Comparisons between our GPR derived estimates of depth, density, and SWE are summarized in Table 2 and in lines 434 to 442 of the revised manuscript. Aside from Line 3, which was located 1.5 meters off of Pit 2 and measured a shallower snowpack than observed in the pit, our GPR derived density and SWE estimates agree with manual measurements within 13% for density and 8% for SWE.

*In terms of processes, simplification, the adaptation of Fomel's seismic approach (Line 133) and the NSE metric (Line 477) are gratuitously complicated, fail due to data quality issues (Line 230), and are not justified given the limited data set. There is no case made that the claims of efficiency (e.g. Line 221, Line 508) address actual measurement and processing bottlenecks or are even accurate. Further, once implemented these techniques still require frequent and substantial manual interventions (Lines 230, 377, 447, 460, and 462) culminating in the authors' suggestion that wavelength scale "point diffractors suitable for this type of analysis can be scouted for ahead of time during summer months or on aerial photographs" (Lines 499). Traditional GPR methods are much simpler than this (e.g. do not require advance scouting or aerial observation).*

The results presented in our revised manuscript include six field data sets, 4 snow density pits, and 86 probed depth measurements. Utilizing our revised processing flow, we get similar results to those presented in the original manuscript, but with more data and without intervention.

Our results are summarize in Table 2 and in lines 700-708 of the revised manuscript:

"To validate the method, we compared estimated snow densities, depths, and SWE to observations made in four snow pits and to 86 probed snow depth measurements. The results are summarized in Table 2 and in Figure 9. If we exclude the two obvious outliers (Figure 10a), the RMS error for our depth predictions for the remaining 88 depth observations is 12% of the mean snowdepth observation. The RMS error for snow density and SWE relative to the mean observed values are 15% and 18%. Averaging the velocities across the entire line (Figure 10 red crosses) reduce the difference between predicted and observed depth values to an RMS error of 9%, suggesting that lateral variations in snow velocity are minimal. Averaging the velocities across the entire line reduces the RMS errors for density and SWE to 8% and 10%, respectively. "

We have removed the statement about scouting for point diffractors and instead comment on the high likelihood of encountering such objects in mountain watersheds (Lines 755-761):

"The data presented in this paper contained an abundance of diffractions located near the soil/ground interface allowing an average velocity for the entire snowpack to be obtained. These events are likely due to small-scale variations in surface topography, rocks, and/or vegetation along the ground surface, which may not be present in all environments. However, we note that mountain watersheds free of vegetation, small undulations in surface topography, and surface rocks are probably rare. Thus, the method may be useful in many regions where seasonal snowpacks exist."

*In terms of reliable SWE estimation, the authors show that the presented approach "does not allow us to confidently differentiate between dry snow and moist snow" (Line 350). Further, authors show that the collected data does not exhibit the frequency dependent attenuation upon which the entire approach depend (Line 318 and Line 331) and even though "within uncertainty bounds there is no resolvable frequency change" (Line 440) the authors go on to use that uncertainty speculate that "there may be up to a 36 MHz shift" and then estimate a volumetric water content from it (Line 442) even though nothing suggests the existence of such a shift beyond the authors' assumption and assertion it should exist (Line 318). This is inappropriate and, as written, likely invalidates the results and conclusions of the paper. Further, the lack of observed frequency dependence is likely due to an actual lack of frequency dependence in radar attenuation in ice for frequencies below 1 GHz (Gudmandsen, 1971). This must be addressed. At the very least, the authors' statement "averaging mean snow-densities from manual observations may be a better strategy" (Line 489) makes it clear that the stated goal to "obtain reliable SWE estimates" (line 144) was not met.*

We apologize for the confusion regarding radar attenuation in wet snow and its relationship to volumetric water content. We agree that in ice (or dry snow) the signal attenuation is negligible. It is the presence of water in the snowpack and the high contrast between the dielectric properties of dry snow and water that cause the attenuation of the GPR signal (see Bradford et al., 2009). Since the attenuation coefficient radar waves propagating in water is a linear function of frequency (Turner and Siggins, 1994), we expect to see a systematic decrease in higher frequency energy that is represented by a systematic decrease in the mean frequency content of the signal.

We recognize that our discussion of radar signal attenuation was a distraction from the main point of our paper and we have reduced the discussion of signal attenuation to one paragraph and moved it to **Section 2.7 Estimating SWE**. In addition, we have not used this analysis to report the maximum volumetric water content within our data resolution.

The revised discussion of radar attenuation (Lines 417-432) reads:

"In this paper, we are primarily concerned with measuring radar velocities and we assume that our data measure the properties of dry snow. The real part of the dielectric constant for water (~80) is much larger than that of snow (~1.5 - 2) and the imaginary part, which describes the attenuation of the signal, is non-negligible (Bradford at al., 2009). The dry snow assumption can be tested from the data by analyzing the attenuation properties of the snowpack (Bradford et al., 2009). The attenuation coefficient for radar waves in water is frequency-dependent (i.e. Turner and Siggins, 1994), with the higher frequencies attenuating more rapidly that the lower frequencies because they go through more cycles per distance traveled. When liquid water is present in the snow, the ground reflection will have a lower mean frequency content than a reference event (the snow reflection for the snowmobile collected data and the direct arrival for the skier-pulled data). To test the dry snow assumption, we calculate the maximum local instantaneous frequency (Fomel, 2007) within a time window surrounding the event of interest then average this value across all of the traces in the GPR image. The standard deviation provides an estimate of the measurement uncertainty. We note that at 500 MHz, a small shift in frequencies results in a non-negligible volumetric water content.."

**Technical Corrections:**

*Line 107: Add an explanation and citation for "targets that have lateral dimensions that approximate the wavelength of the signal".*

The diffraction literature makes reference to objects that are of similar dimension to the signal wavelength as well as to the first Fresnel zone. In consideration of our analysis depicted in Figure 1, we suggest that the Fresnel zone is more appropriate for our purposes. Lines 129-132 read:

"In this paper, we are specifically interested in targets that have lateral dimensions that are less than the Fresnel zone. These objects scatter energy in all directions and appear on the raw GPR image as hyperbolic events, called diffractions (Landa and Keydar, 1998)."

Citation:

Landa, E. and S. Keydar.: Seismic monitoring of diffraction images for detection of local heterogeneities, *Geophysics*, 63, 1998.

*Lines 122: Why mention the 800 MHz data if it's not used in the analysis? Consider removing.*

We agree with this comment and the revised manuscript does not mention the 800 MHz data.

*Line 202: This entire paragraph has a level of detail that seems inappropriate for a journal manuscript. Consider dropping or revising.*

We think that this paragraph may be useful for readers who are unfamiliar with migration and have kept it in the revised manuscript.

*Line 235: Provide a justification for the "equally likely" claim.*

Thank you for this comment. We have justified our uncertainty estimate by comparing images migrated at different velocities to the corresponding varimax value. We found that images that were visually indistinguishable corresponded to varimax values that were greater than approximately 95% of the maximum. (See Figure 2 of the revised manuscript). Because the migrated images were indistinguishable to the human eye, we interpret these migration velocities to be valid velocity estimates and use them as upper and lower bounds on the true velocity.

*Line 251: The units of the left and right side of this equation do not match.*

Thank you for pointing out this error. The equation should be:

$$V_{snow} = \sqrt{\frac{V_{mig}^2 t_{soil} - V_{air}^2 t_{snow}}{t_{soil} - t_{snow}}}$$

and we have fixed this in the revised manuscript.

*Line 318: Add an explanation and citation for "the coefficient increases with increased frequency" and address the fact that, in ice and below 1 GHz it does not (Gudmandsen, 1971).*

We have added a citation and explanation. In the revised manuscript Lines 422-432 read:

"The attenuation coefficient for radar waves in water is frequency-dependent (i.e. Turner and Siggins, 1994), with the higher frequencies attenuating more rapidly that the lower frequencies because they go through more cycles per distance traveled. When liquid water is present in the snow, the ground reflection will have a lower mean frequency content than a reference event (the snow reflection for the snowmobile collected data and the direct arrival for the skier-pulled data). To test the dry snow assumption, we calculate the maximum local instantaneous frequency (Fomel, 2007) within a time window surrounding the event of interest then average this value across all of the traces in the GPR image. The standard deviation provides an estimate of the measurement uncertainty. We note that at 500 MHz, a small shift in frequencies results in a non-negligible volumetric water content."

Citation:

Turner, G., and A. F. Siggins.: Constant Q attenuation of subsurface radar pulses, *Geophysics*, 59, 1994.

*Line 155, 404, and others: Justify uncertainties and provide basis throughout the manuscript for uncertainties that are currently just asserted.*

Lines 341-343:

"We use the upper and lower bounds on our velocity estimates to compute upper and lower bounds on all subsequent calculations."

Lines 401-403

"To propagate our velocity uncertainty estimates through the dix equation, we assign a travel-time uncertainty of 0.2 ns to our travel-time observations and use Eq. 5 along with our velocity uncertainty estimates to compute upper and lower bounds on the snow velocity."

**Reviewer # 2**

*The paper by St. Clair and Holbrook describes a novel approach for determining the snow-water-equivalent (SWE). I have read the well-written paper with great interest, and I judge it a useful contribution for the Cryosphere community. Before the paper can be accepted, I suggest that the authors address the following issues.*

*1. There are inconsistencies and errors with the units. I suggest that they should use mks units throughout the entire manuscript.*

In the revised manuscript we use meters for units of distance (and depth) and SWE, m/ns for radar velocity and $kg/m^3$ for snow density.

*2. It is often referred to Line 7 and Line 19. Without a map showing these profiles and/or a table describing the characteristics of the profiles, these references are not helpful. Since there seem to be only lines 7 and 19 discussed, it would make sense to rename them to line 1 and line 2.*

We have provided tables to describe each field data set. Table 1 summarizes snowpits 1-4. Table 2 describes the GPR data and reports data of acquisition, the acquisition mode (ski or snowmobile), the type of independent observations we compared our results to, and the difference between GPR estimated values and manually measured values.

*3. It is my understanding that the methodology works only for dry snow and that de- termining the liquid water content from the GPR data was not successful. These two facts should be stated more explicitly.*

We have made this more explicit in several parts of the revised manuscript:

Lines 99-101:

"Since our primary goal is to develop a method for quick velocity estimations, we assume that the snow we are measuring is dry."

Lines 417-418;

"In this paper, we are primarily concerned with measuring radar velocities and we assume that our data measure the properties of dry snow."

*4. Figures 10 and 11 indicate that the GPR estimated depths are systematically smaller compared with the probe depths. This should be discussed.*

After reprocessing the data with the new approach, there is not a systematic error in the depth predictions.

*5. Further discussion is required on the basic assumption of the approach that the scattering bodies can be considered as a point diffractor. I guess that the scattering occurs mostly at sizeable boulders. Therefore, I am not convinced that the point scatterer assumption is really justified. The diffraction hyperbola of a finite sized scattering body is expected to appear wider in a GPR profile, which would likely result in an over- estimation of the snow velocity. Consequently, one would thus rather overestimate the snow depth, but, as mentioned in point 4, the opposite seems to be the case.*

Thank you for this comment. We have evaluated the effects of non-finite point diffractors, or diffracting objects that approach the dimension of the Fresnel zone. Figure 1 illustrates the effects of a lateral diffractor dimension and curvature on the peak value of the varimax norm. We also include the following discussion:

Lines 286-305:

"To assess possible errors in the migration velocity analysis, we applied our workflow to a synthetic data set generated from diffractors of varying size. The Fresnel radius is given by $R_f = \sqrt{\frac{z\lambda}{2}}$ (Sheriff, 1980) where z is depth and $\lambda$ is the dominant wavelength. Figure 1 shows the effect of such an event on V. We created five synthetic diffractions with migration a migration velocity of 0.24 m/ns. The first four (Figure 1a) correspond to rectangular objects at 1 meter depth with horizontal dimensions 0.1, 0.2, 0.3 and 0.4 meters, and thickness of 0.03 m and the fifth corresponds to a circular object with a radius of 0.4 meters (close to $R_f$ for the 500 MHz ricker wavelet used to generate the diffractions). The corresponding varimax curves for the windows shown in Figure 1a are plotted in Figure 1b. The V curves are peaked at 0.24 m/ns for all of the rectangular diffractors, with flatter (less well-resolved) peaks as the horizontal dimension of the diffracting object increases, suggesting a larger uncertainty in the velocity estimate. The peak V value for the circular diffractor is at 0.268 m/ns, indicating that curved objects with lateral dimensions close to the size of the Fresnel zone may continue to focus at velocities higher than their true velocity. Finally, Figure 1c shows the V curve for the entire image, peaked at the correct velocity of 0.24 m/ns. This analysis suggests that the peak V value will correspond to the correct velocity if the majority of the diffractions correspond to objects much than $R_f$..”

Lines 709-726:
“The greatest potential for systematic error in this analysis is the presence diffracting objects whose dimensions exceed the radius of the first Fresnel zone. The field data offer the opportunity to evaluate the influence of diffractor size on velocity estimates. Line 1, for example, shows four prominent diffractions between 50 and 70 meters. The Varimax norm has a maximum value at 0.256 m/ns, which is the velocity that focuses the two leftmost diffractions (Figure 6c). The diffractions on the right are clearly not focused because they are caused by an object (most likely a log) with a radius greater than the first Fresnel zone. Because the leftmost two have a higher amplitude then the others, they have the largest influence on the varimax value. Thus, although there are clearly events in the field data that have the potential to give erroneous results, our results suggest that reliable velocity estimates can be achieved so long as the majority of the diffracted energy is related to objects that can be considered point diffractors.
.”

*6. Some of the figures are lacking proper axis labeling.*

In our original manuscript many of the plots did not have axes labels, if the plots immediately above or adjacent to them had the same axis. The lower and leftmost plots had the axes labels on them. In the revised manuscript, all plots have axes labels.

*7. It seems that in Figures 7 and 9 only portions of the GPR profiles are shown. This should be clarified.*

In the revised manuscript, all figures depicting data show the entire data set.

[revised manuscript text omitted]

... [19]

**Figure 4**

[Figure]

**Figure 5**

[Figure]

**Figure 6**

**Figure 7**

[Figure]

**Figure 8**

**Figure 9**

[Figure]

**Figure 10**

[Figure]

**Tables**

**Table 1. Snowpit summary**

| Pit Name | Date | Depth (m) | Rho (kg/m$^3$) | SWE (m) | GPR profiles |
|---|---|---|---|---|---|
| Pit 1 | 25-Feb-15 | 1.33 +/- 0.05 | 305 +/- 44 | 0.40 +/- 0.14 | Line 1 |
| Pit 2 | 26-Feb-15 | 1.56 +/- 0.05 | 314 +/- 44 | 0.49 +/- 0.14 | Lines 2 and 3 |
| Pit 3 | 11-Mar-15 | 1.44 +/- 0.05 | 379 +/- 50 | 0.55 +/- 0.13 | Line 4 |
| Pit 4 | 11-Mar-15 | 1.80 +/- 0.05 | 360 +/-48 | 0.65 +/- 0.13 | Line 4 |

**Table 2. Summary of GPR field data and comparison to manual measurements**

| GPR Profile | Collection Date | Acquisiton Mode | Pits/Probe | GPR Predictions at pit | | | Error Compared to Pit/Probe | | |
|---|---|---|---|---|---|---|---|---|---|
| | | | | Depth Pred (m) | Rho (kg/m$^3$) | SWE (m) | Depth | Rho | SWE |
| Line 1 | 25-Feb-15 | Ski | Pit 1 | 1.29 +/-0.06 | 288 +/- 50 | 0.37 +/- 0.07 | 2.6% | 5.5% | 8.0% |
| †Line 2 | 25-Feb-15 | Ski | Pit 2 | 1.59 +/- 0.04 | 294+/-40 | 0.46+/- 0.03 | 0.1% | 6.0% | 6.0% |
| †Line 3 | 25-Feb-15 | Snowmobile | Pit 2 | 1.10 +/- 0.05 | 354 +/-65 | 0.39 +/ 0.06 | *30.0% | 13% | *21% |
| Line 4 | 11-Mar-15 | Snowmobile | Pit 3 Pit 4 Probes | 1.50 +/- 0.08 1.91 +/- 0.12 | 389 +/-92 394 +/- 97 | 0.53+/- 0.09 0.69 +/- 0.13 | 6.0% 6.0% **RMSE = 0.13 m (9%) | 3% 10% | 2.0% 6.% |
| †Line 5 | 17-Mar-15 | Snowmobile | Probes | | | | **RMSE = 0.38 m (18%) | | |
| †Line 6 | 17-Mar-15 | Snowmobile | Probes | | | | **RMSE = 0.19 m (11%) | | |

*Line 3 was located 1.5 meters off of Pit 2, disagreement between depth and SWE measurements at this site reflect lateral variations in snowdepth.

**RMSE percentages are calculated relative to the mean observed depth along each profile

†Lines 2, 3, 5, and 6 are described in the supplementary materials.

**Supplementary Materials**

In addition to the data presented in the main text, we include snow density profiles for Pits 1-4 (Figure S1) and show data and results for GPR lines 2, 3, 5 and 6:

Line 2 is a skier pulled data set collected on February 25, 2015 in sub-freezing conditions. Pit 2 was located at x = 100 meters. After interpolating the data to equal spacing, the trace spacing was 0.027 meters. The data and velocity picks are depicted in Fig S2 and the resulting snow depth, density and SWE estimates are shown in Fig S3.

Line 3 is a snowmobile driven data set collected on February 25, 2015 in sub-freezing conditions. Pit 2 was located at x = 54 meters. After interpolating the data to equal spacing, the trace spacing was 0.027 meters. The data and velocity picks are depicted in Fig S4 and the resulting snow depth, density and SWE estimates are shown in Fig S5. Notably, Pit 2 was located ~1.5 meters off of the GPR line, which we suggest explains the discrepancy in the depth and SWE predictions at the pit site.

Line 5 is a snowmobile driven data set collected on March 17, 2015 in above-freezing conditions. Air temperature reached 10° C on this day and we infer that the dry snow assumption was not valid. After interpolating the data to equal spacing, the trace spacing was 0.0245 meters. The data and velocity picks are depicted in Fig S6 and the resulting snow depth, density and SWE estimates are shown in Fig S7. We probed snowdepth along this line at 2 meter intervals.

Line 5 is a snowmobile driven data set collected on March 17, 2015 in above-freezing conditions. Air temperature reached 10° C on this day and we infer that the dry snow assumption was not valid. After interpolating the data to equal spacing, the trace spacing was 0.0148 meters. The data and velocity picks are depicted in Fig S8 and the resulting snow depth, density and SWE estimates are shown in Fig S9. We probed snowdepth along this line at 2 meter intervals.

[Figure]

**Figure S1.** Snow density profiles. Black lines are measured density values, red lines indicate uncertainty estimate.

[Figure]

**Figure S2. a)** raw GPR data for Line, red line indicates interpreted ground reflection 2 **b)** GPR data after PWD filtering **c)** diffractions migrated at the mean velocity (0.243 m/ns) for the entire line **d)** Normalized varimax curves for sliding window 10 meters wide. Blue curve shows the peak value for every curve, red line is smoothed with a box car averaging filter 10 meters wide.

[Figure]

**Figure S3.** Line 2 Results. **a)** density, **b)** snow depth (black line) and SWE (blue line) estimates from the GPR data, snow pit data are shown in red. Grayed out region corresponds to areas where velocity picks are unreliable.

[Figure]

**Figure S4. a)** raw GPR data for Line 3, red lines indicates interpreted ground and snow reflections **b)** GPR data after PWD filtering **c)** diffractions migrated at the mean velocity (0.247 m/ns) for the entire line **d)** Normalized varimax curves for sliding window 10 meters wide. Blue curve shows the peak value for every curve, red line is smoothed with a box car averaging filter 10 meters wide.

[Figure]

**Figure S5.** Line 3 Results. **a)** density, **b)** snow depth (black line) and SWE (blue line) estimates from the GPR data, snow pit data are shown in red. Grayed out region corresponds to areas where velocity picks are unreliable.

[Figure]

**Figure S6.  a)** raw GPR data for Line 5, red lines indicates interpreted ground and snow reflections **b)** GPR data after PWD filtering **c)** diffractions migrated at the mean velocity (0.233 m/ns) for the entire line **d)** Normalized varimax curves for sliding window 10 meters wide. Blue curve shows the peak value for every curve, red line is smoothed with a box car averaging filter 10 meters wide.

[Figure]

**Figure S7.** Line 5 Results. **a)** density,  **b)** snow depth (black line) and SWE (blue line) estimates from the GPR data, snow pit data are shown in red. Grayed out region corresponds to areas where velocity picks are unreliable.

[Figure]

**Figure S8.** **a)** raw GPR data for Line 6, red lines indicates interpreted ground and snow reflections **b)** GPR data after PWD filtering **c)** diffractions migrated at the mean velocity (0.245 m/ns) for the entire line **d)** Normalized varimax curves for sliding window 10 meters wide. Blue curve shows the peak value for every curve, red line is smoothed with a box car averaging filter 10 meters wide.

[Figure]

**Figure S9.** Line 6 Results. **a)** density, **b)** snow depth (black line) and SWE (blue line) estimates from the GPR data, snow pit data are shown in red.